# Socioeconomic and Environmental Impact Assessment of Different Power-Sourced Drip Irrigation Systems in Punjab, Pakistan

Iftkhar Ul Hussan [1], Muhammad Nadeem [2,*], Muhammad Yamin [2], Sikandar Ali [3], Muhammad Mubashar Omar [4], Shaheer Ahmad [2], Mamoona Zulfiqar [5] and Tallat Mahmood [1]

[1] Department of Agriculture (Water Management), Government of Punjab, Lahore 54700, Pakistan
[2] Department of Farm Machinery and Power, University of Agriculture, Faisalabad 38000, Pakistan
[3] Agricultural Remote Sensing Lab, University of Agriculture, Faisalabad 38000, Pakistan
[4] Department of Energy Systems Engineering, University of Agriculture, Faisalabad 38000, Pakistan
[5] Centre of Excellence in Water Resources Engineering, University of Engineering and Technology, Lahore 54700, Pakistan
* Correspondence: ndmmuhammad@dal.ca; Tel.: +92-308-000-6868

**Abstract:** This research investigated the best economically viable power source with the least environmental impact and socially acceptable for the maize crop. Maize is one of the key economic crops in Pakistan. Solar-, electric-, and diesel-powered drip irrigation systems (DIS) were considered for comparative study. We selected 45 sites of maize crop to collect the data, with an area of 1–3 ha, from three divisions. For economic viability, the benefit:cost ratio, life cycle cost, and payback period were calculated, and $CO_2$ emissions were calculated to assess the environmental impact. The SPSS model was used for one-way ANOVA followed by post hoc and chi-squared tests to check the significance level between all power sources. It was found that the B-C of electric power, solar, and diesel drip irrigation systems was 1.65, 1.52, and 1.44, respectively. Solar, diesel, and electricity power DIS have $CO_2$ emissions of 0.02, 0.730, and 1.106 tons/ha, respectively. The research concludes that solar power and electric power are the best sources for the environment and economically, respectively. It is recommended that solar power DIS be subsidized, which will help to lower $CO_2$ emissions and reduce the electricity shortfall in Pakistan.

**Keywords:** benefit:cost ratio; DIS; high-efficiency irrigation system; irrigation systems; solar energy; alternate energy resources

## 1. Introduction

Pakistan possesses one of the world's largest irrigation infrastructures, with three major dams with design capacities of over 20 BCM (billion cubic meters), 23 barrages, headworks, and 45 main canals dominating an area of approximately 17 Mha (million hectares) [1]. Agriculture serves as the foundation of the economy in a country that uses more than 93% of its natural water resources [2]. Agribusiness is Pakistan's most critical division and the foundation of the economy, whose efficiency relies upon various rural and common assets: 18.9% of Pakistan's GDP is in the farming area, which utilizes 42.3% of the labor [3]. However, there are a number of difficulties with this method. Water scarcity is considered one of the biggest challenges [1]. The Falkenmark indicator establishes a link between accessible water supply and the human population in water-stressed countries, and according to it, Pakistan has per capita water of less than 1700 m$^3$. A nation whose per capita water accessibility drops below 1000 m$^3$ is classified as water-scarce. However, a country suffers absolute water scarcity when per capita water drops below 500 m$^3$ [4]. In 1950, Pakistan had a per capita water accessibility of around 5000 m$^3$. In 1990, Pakistan reached the water-stress limit, and in

2005, Pakistan reached the water-scarcity limit. If the current trend continues, this will reach the extreme water-scarcity threshold by 2025 [1].

The Water Resources Vulnerability Index (WRVT) [5] relates the yearly national availability of water to the total yearly consumption. The country is defined as being water-limited if yearly withdrawals are between 0 and 40% of the annual water supply. If it surpasses 40%, the region is deemed to be extremely water-stressed. Pakistan has a WRVT of around 77%. In 2004, the shortfall of water was 11%, which is estimated to increase to 31% by 2025 [6]. Because of water scarcity and population growth, there will be a food deficit of around 70 million tons by 2025 [7]. The performance level is less than 40% due to inadequate irrigation system management. For example, in 2001 the government of Pakistan reported that only 55 BCM of the 142 BCM of available water at canal headworks is consumed by crops [6]. The leftover 87 BCM (61%) is wasted during transportation and application in the field. Water lost because of seepage in regions underlain by groundwater sources is just a temporary loss since it may be retrieved when necessary. However, because power is consumed to pump this water, it represents an energy loss. This is a permanent loss in places underlain by salty groundwater since it cannot be utilized for irrigation or drinking and may cause waterlogging and salinity [1,8].

Compared to India and California (1.5 kg/m$^3$), Pakistan has relatively low water productivity (0.6 kg/m$^3$) [9]. To address the problem of food security, water production should be increased by implementing various efficient irrigation techniques. In 2002, Allah Bakhsh and Awan reported that Pakistan is located in a low precipitation area, with rainfall ranging from less than 100 mm in the lower Indus plains to much more than 750 mm in the northern areas, which is below crop water demands in various regions, such as 1400 mm in Balochistan (Turbat), 1280 mm in Punjab (Faisalabad), 900 mm in Khyber Pakhtunkhwa (Parachinar) and 1487 mm in Sindh (Jacobabad) [10]. However, water-saving irrigation methods are required to meet agricultural water requirements. Modern irrigation systems are more important than any other public development in arid and semiarid climates [11,12]. Water is delivered practically directly to each plant via a pipe system, and the field area surrounding every plant is wet. Water is delivered to the plant on a low-tension and frequent basis, allowing for very high water usage efficiency [12]. Microirrigation systems are more advantageous and viable in locations where irrigation is ineffective or land reformation requires large sums of money [12].

Pakistan possesses one of the world's biggest groundwater resources. The number of tube wells climbed from fewer than 0.2 million in 1980 to 1.1 million in 2015 [13]. This is mostly owing to increasing crop productivity, which climbed from 67% to roughly 150% during the same time frame [14]. The government has not imposed any restrictions on the installation of agricultural tube wells, and growers have taken advantage of this availability to farm as much land as possible [15]. According to research studies, using the drip technique saves 40–70% of the water plus increases output by 10–100% for certain crops [16,17].

Diesel-powered pumps are often placed at modest depths of 20 to 40 ft. The average land size with diesel tube wells is below 5 ha, accounting for about 85% of the whole. According to current projections, diesel-powered tube wells emit roughly 5.025 million metric tons of $CO_2$ each year [18]. Solar-powered drip irrigation is a reasonable alternative to diesel-powered tube wells, but widespread adoption in Pakistan has been fraught with difficulties [15]. Punjab is also Pakistan's most inhabited province, and thus it will be useful to investigate the distribution of both diesel and electric pumps well throughout the region. In 2014, the Agriculture Department of Punjab reported that 87% of the tube wells are diesel-driven, with the remaining 13% being electrically operated [15,19].

In 2012, Pakistan's electricity mix comprised hydroelectric, thermal, and nuclear power facilities. Approximately 28.40% of electricity was generated by hydropower facilities, 67.82% by thermal power stations, and the remaining 3.78% by nuclear and renewable energy generation systems [20]. In 2014, Pakistan's current capacity installed for the power grid supply was 23,644 MW. The state's suppressed electricity requirement is 19,735 MW in the summer and 14,922 MW in the winter [21]. Despite the installation of almost 3000 MW in the previous five years, the supply had not really progressed to the point where it could meet the energy demand of the current connected load [21,22]. In May 2011, the Pakistan Electric Power Company reported an electrical deficit of around 7000 megawatts (MW) [23,24]. In 2012, Aafia Malik wrote that there was a 3000–5000 megawatt demand–supply mismatch, leading to a 4–12 h blackout in the country's urban and rural areas [21,22]. Because of the extensive use of fossil fuels, the energy production industry contributes significantly to $CO_{2eq}$ emissions. The energy industry accounts for approximately 76.1% of total $CO_{2eq}$ emissions in Pakistan [25]. Between 2007 and 2017, the market for petroleum products climbed at a rate of 4% each year. Overall petroleum consumption has been dropping since 2018 due to a reduction in the use of furnace oil in the power industry. As of 2019, transportation is the largest user of petroleum products, accounting for approximately 76.6%, followed by energy production, which accounts for approximately 14%, and industry, which accounts for approximately 7.6%, with the remainder being shared by residences, agricultural sectors, and other governmental agencies [26]. The primary obstacles to implementing an improved irrigation system in the country are the rise in fuel prices and the existing breakdown of power.

Although many studies have been conducted on the use of solar power drip irrigation systems around the world, this invention is still in its early stages in Pakistan. This paper not only focuses on the economic analysis of solar-coupled high-efficiency irrigation systems with the comparison of diesel and electric power drip irrigation systems but also discusses the social adoption and environmental impacts of all power sources for drip irrigation systems. The objective is to evaluate and compare the economic viability of different power sources for drip irrigation systems in Punjab, assess the environmental impacts of different power sources for drip irrigation systems in Punjab, Pakistan and assess the social adoption of different power sources for drip irrigation systems in Punjab. The hypothesis of this study was that a solar system is an economically suitable energy system for drip irrigation in Pakistan.

## 2. Materials and Methods

### 2.1. Description of Study Area

Previous research reported higher produce yield, lower weed yield, improved germination and water saving using drip irrigation [27–29]. Individual regions of Pakistan have different sorts and volumes of assets, for example, farmland, land, water, domesticated animals, horticultural credit, rural hardware, manure, and farmer specialists. Three divisions selected for this study: Faisalabad, Sahiwal, and Lahore (Figure 1). The coordinates for these areas are shown below in Figure 1.

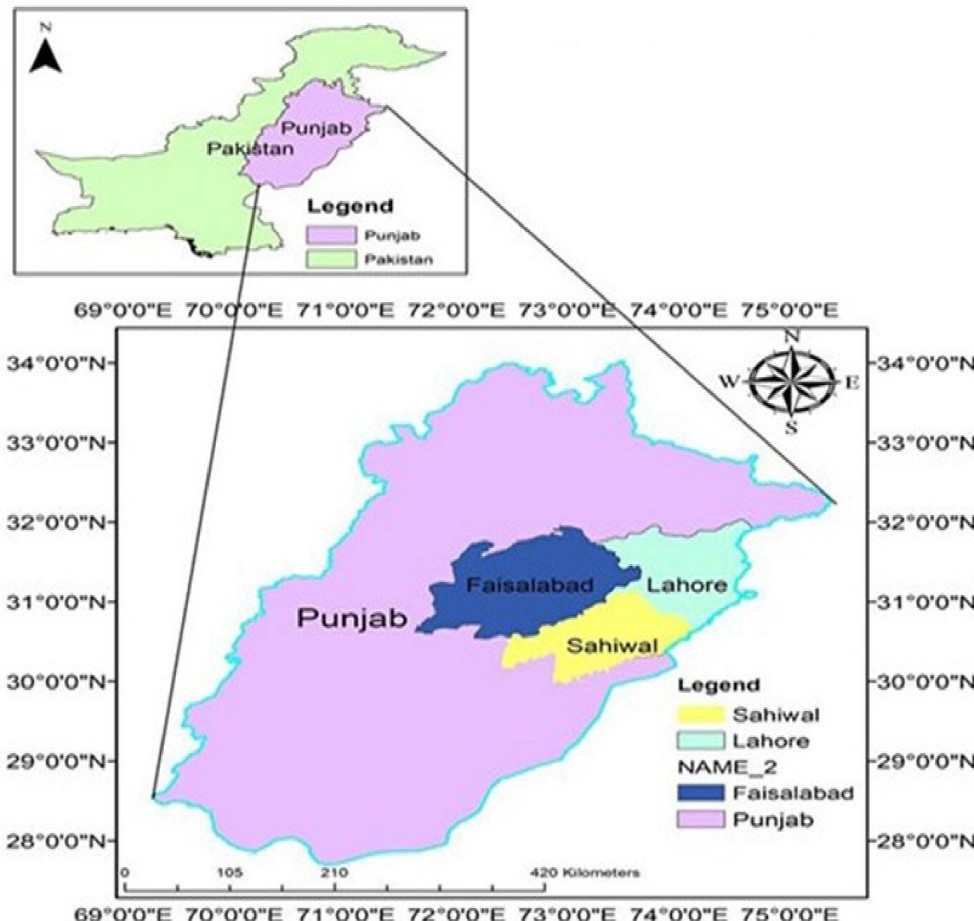

**Figure 1.** Location maps of Punjab.

*2.2. Data Collection*

This section provides information on the required data to conduct the research study. All data for maize crops on drip irrigation systems with different power sources from Punjab were collected. Purposive sampling technique was used to collect and analyze the data.

Sampling of Drip Irrigation System (DIS) Sites

The baseline survey and impact assessment data collections for completed drip irrigation system sites were conducted. For this purpose, based on stratified sampling, 45 DIS sites were selected out of 139 sites installed by the Punjab government during the last five years. Data were gathered from across Punjab. The selection of sites was random to ensure the quality of the data as well as covering the whole sites. Figure 2. shows site selection dependent on drip irrigation-installed area between the limits of 1 ha to 3 ha, 3 ha to 5 ha, and 5 ha to 7 ha, and further sites dependent on power sources on an equal basis as solar, diesel, and electricity. However, this study focused on small farms with area 1–3 ha and all sampling was done in the study area.

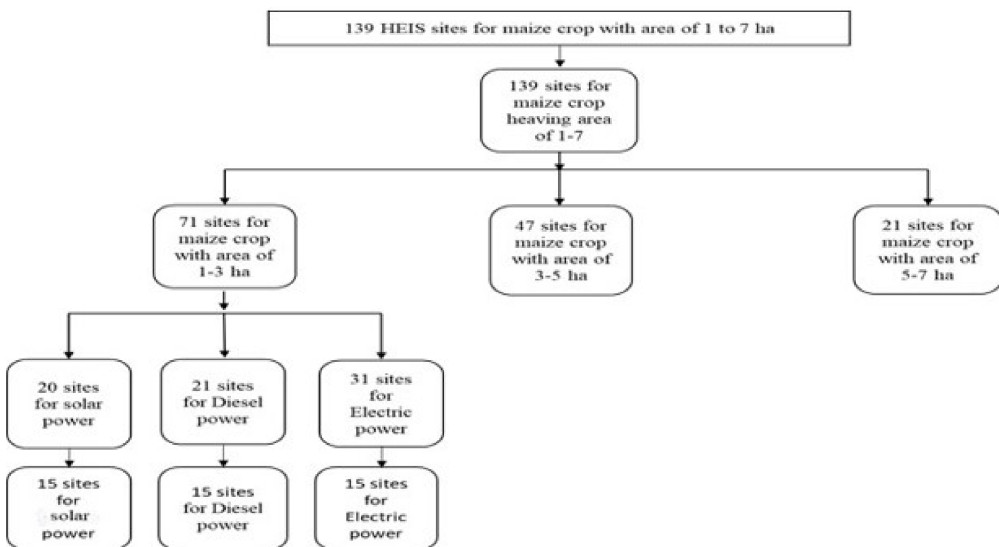

**Figure 2.** Selection of HEIS site framework.

Total HEIS in Punjab is further divided by farm size, such as large farm sizes ranging from 5–7 ha, medium 3–5 ha, and small 1–3 ha, respectively. We checked the data availability and number of sites before the decision on area and then decided to evaluate all parameters within the limits of a small farm size, which is 1–3 ha. The selected sites in the Faisalabad, Lahore, and Sahiwal divisions were further arranged by their power source with an area limit of 1–3 ha. A sample size of 15 for each source was selected, for a total of 45 sites selected for the study.

After the collection of data from the On Farm Water Management Department (OFWM), we compiled a list of farmers from each division: Faisalabad, Sahiwal, and Lahore. The interest of the farmer also changes with landholding capacity, that is, whether he is the owner of 100 ha or 5 ha. While selecting the 45 sites, all the technical parameters were considered, such as area, soil type, cropping pattern, etc. Crop was the most important parameter. The selected crop was maize, and all the data were collected for maize only. Diesel engines are less efficient and lift water from smaller heads that vary from 6 to 15 m. Diesel engines do not face transmission and distribution losses, whereas in the case of electric power irrigation systems, they face all these losses. DIS sites are cited in individual divisions, as 45% of the selected sites are situated in Sahiwal division and only 22% in Lahore, whereas the rest are in Faisalabad division.

Most of the data regarding DIS sites and total cost incurred were collected from the On Farm Water Management office in Lahore, whereas the crop data, such as all inputs from sowing till harvesting, were collected from the farmer directly via an author survey. The output cost data Were also collected. A schematic from data collection to analysis is shown in Figure 3 below. Basically, the overall methodology can be categorized into two main parts:

1.  Data collection.
2.  Data analysis.

The methodology is further divided into steps, such as data collection and analysis while adopting the models or methods to get the results.

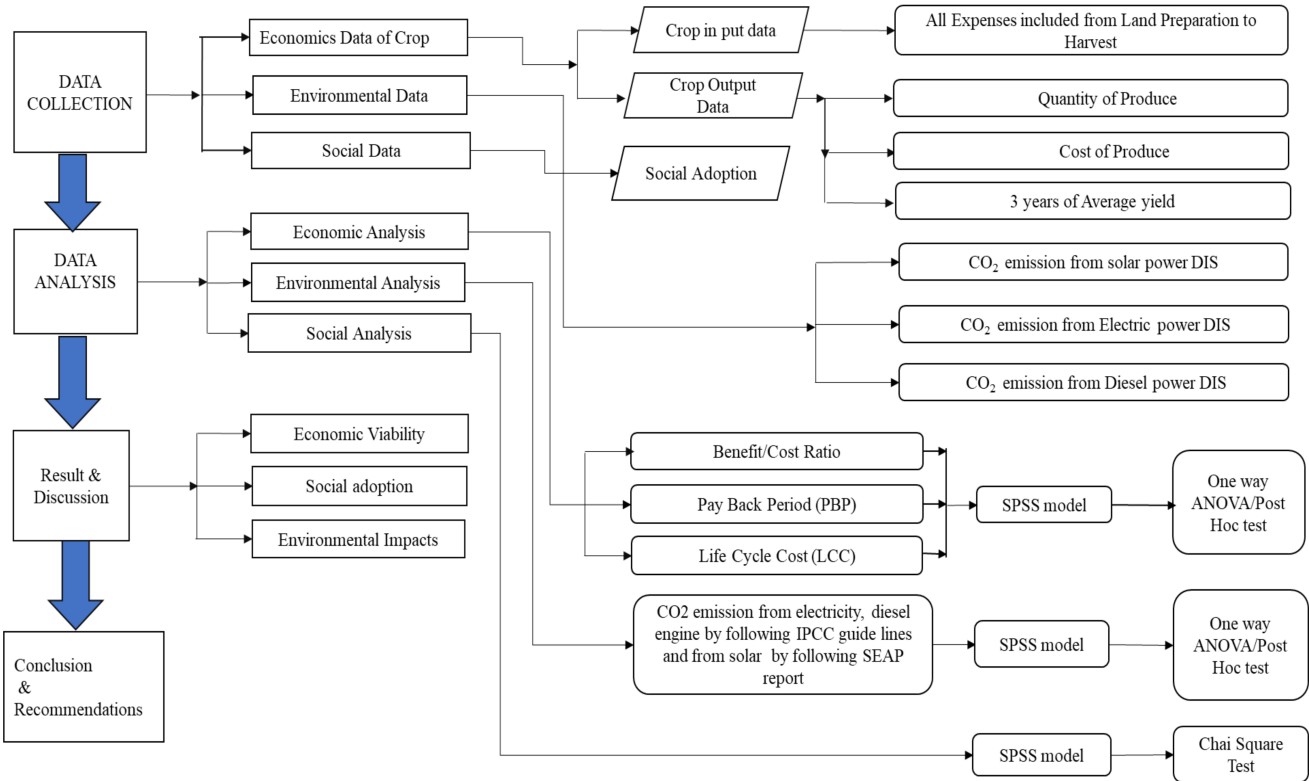

**Figure 3.** Overall methodological framework.

### 2.3. Economic Data

### 2.3.1. Crop Input and Output Data

Nowadays, most of our farmers maintain records of crop input data as well as crop output data. This information expresses farm yield, which we need for economic analysis. As such, a primary farm survey is more useful and can be done easily. In this research, the primary data were collected from farmers for economic analysis and social adoptability. The whole dataset was collected for maize.

### 2.3.2. Environmental Data

#### $CO_2$ Emission

In the IPCC guidelines, we calculated the emission factor of $CO_2$ for all provinces of Pakistan. They have calculated values for $CO_2$ emissions for diesel too. Both the values were taken and be used for the calculation of $CO_2$ emissions at each HEIS site, whether an electric power or diesel power irrigation site. In the SEAP guidelines, the emission factor for $CO_2$ from locally used solar power systems was calculated, which was used to calculate the emissions from the solar power drip irrigation system. Human activities release more than 29 billion tons of $CO_2$ into the environment every year [30]. Pakistan is expending to over 98% of nonrenewable (fossil-filled) vitality, which is contaminating the earth [31]. For screening greenhouse gases and air toxins over the sub-landmass, a few investigations connected with various practices, for example, satellite and ground-based remote detection and in situ estimations, have been conducted [32]. Plants also use $CO_2$ to play an important role in the photosynthesis cycle and recover the water proficiency required by plants [33]. $CO_2$ discharges contribute 74% of all $CO_2$ production [34]. Strategy objectives include improving energy utilization and carbon sinks, as well as empowering manageable types of agriculture to combat climate change [35].

Social Adoption Data

We considered the amount of drip irrigation acceptance of farmers by education. Data regarding social adoption for each source were collected from all the sites visited for economic data. We used a questionnaire-based survey to evaluate the farmer's adoption and to know the basic reasons for the adoption of any source by the farmer. In terms of social data, we investigated farmer education, family size, farm operator education, total landholding, farmer interest, interaction with other farming communities, interaction with agriculture and the OFWM department, knowledge of the latest technologies, and so on.

*2.4. Data Analysis*

The whole framework of data analysis for each of the objectives along with model and statistical analysis applied to determine significant differences is presented in Figure 3.

2.4.1. Economic Analysis

Economic analysis is carried out to test the viability of any project or system. A detailed feasibility study is required before starting any project. The feasibility study not only speaks of the prestige of the project but also helps in determining the sequence of the project procedure [36]. The goal of this economic study was to find the most economically viable source of energy for the drip irrigation system. The following analysis was carried out to test the economic acceptability.

2.4.2. Benefit–Cost Analysis (BCA)

Basically, this is the measurement of all possible profits and benefits from a project while taking into account the qualitative and quantitative factors. It is also called cost–benefit analysis or benefit:cost ratio (BCR). The BCR is calculated as the present value (PV) of benefits divided by the present value (PV) of costs [37].

$$\text{BCR} = \frac{\sum_{t=0}^{T} \frac{B_t}{(1+r)^t}}{\sum_{t=0}^{T} \frac{C_t}{(1+r)^t}} \tag{1}$$

The current time is t = 0. If the benefit-to-cost ratio is greater than one, then the project is good and can be accepted.

2.4.3. Present Value (PV)

The notion of present value asserts that an amount of cash currently is worth greater than the exact quantity in the future. It can be calculated by [38]:

$$\text{PV} = \frac{p_t}{(1+r)^t} \tag{2}$$

where PV is the present value of capital amount, $p_t$ is value of the future amount in time t, r is the discount rate and t is the year in which $p_t$ is realized.

2.4.4. Net Present Value (NPV)

The difference between the current value of cashflow and withdrawals over a time period is known as net present value (NPV). The NPV is the present estimation of all projects' net advantages. The estimation of net advantages in any period is basically the advantage less the expense [39]. It can be calculated by:

$$\text{NPV} = \sum_{t=0}^{T} \frac{R_t}{(1+r)^t} \tag{3}$$

where $R_t$ is cash inflow–outflow at any time t and r is the discount rate.

### 2.4.5. Payback Period (PBP)

The payback period is defined as the time it takes to recover the capital cost of a project [40]. It can be calculated by:

$$\text{PBP} = \frac{\text{Initial Investment}}{\text{Annual Operating cash inflow}}$$
$$\text{or} \tag{4}$$
$$\text{PBP} = \frac{\text{Cost of Project}}{\text{Annual Cash flow}}$$

### 2.4.6. Life Cycle Cost (LCC)

For the adoption of a new system, the life cycle cost is a key point to consider when making initial investments. Life cycle cost is the representation of all costs over the life of the system. It is based on different parameters such as operation and maintenance costs, fuel costs, salvage costs, and the capital cost of the system. It can be calculated by [41]:

$$\text{LCC} = \text{CC} + \text{MC} + \text{FC} + \text{RC} + \text{SC} \tag{5}$$

where CC = capital cost, MC = maintenance cost, FC = fuel cost, RC = replacement cost, and SC = salvage value.

### 2.4.7. Capital Cost

The capital cost of the system was calculated using data provided by the On-Farm Water Management Department in Punjab, Pakistan. The total project cost was divided by the project area to calculate the capital cost for a one-hectare area.

### 2.4.8. Maintenance Cost

The maintenance cost includes all kinds of repair and maintenance of the drip irrigation system, including the power source. It also consists of drip system parts and wear and tear of the power source.

### 2.4.9. Fuel Charges

This comprises price of fuel in the case of a diesel-powered drip irrigation system as well as electricity usage (kWh) in the case of an electricity-powered drip irrigation system. It may also be considered a running cost.

### 2.4.10. Replacement Cost

The replacement cost consists of the amount spent to replace any part of the system during the study.

### 2.4.11. Salvage Value

This is basically the value of the whole drip irrigation system, including the power source, after the completion of the life of that concerned part or the study period, whichever is completed first. The formula for salvage value is as below [42].

$$\text{Salvage Value} = \text{CC} - (\text{I} \times \text{n}) \tag{6}$$

where CC is capital cost, I is depreciation rate, and n is number of years.

### 2.4.12. SPSS Model

SPSS is a Windows program that can be used to enter information, perform analysis, and prepare tables and charts. It is generally used in the social sciences and in the business world. The SPSS Modeler is an IBM information examination and investigation program. It is used to manufacture prescient models and perform other logical assignments. It has a visual interface that enables users to use statistical and information mining calculations

without programming. It is used by specialists for statistical examination. SPSS version 26 was used in this study.

*2.5. Statistical Analysis*

2.5.1. One Way ANOVA

One-way analysis of variance (ANOVA) was used to determine if there were statistically significant differences between the means of two or more independent (unrelated) groups. One-way ANOVA is often accompanied by a special test to check the differences between the two parameters that are significant and how significant. An independent variable has nominal levels or different ordered levels. One-way ANOVA was used for statistical analysis for both environmental and economic analysis, which was further enhanced by the post hoc Tukey HSD test.

2.5.2. Post Hoc Test

ANOVA tests are performed only to confirm the differences between groups (i.e., a statistically significant one-way ANOVA result). A posteriori tests try to check the frequency of incorrect experiments (usually alpha = 0.05). A post hoc test is called an a posteriori test, that is, it is done after the ANOVA.

2.5.3. Environmental Analysis

In this study of drip irrigation systems with different power sources, the most important factor is $CO_2$ emissions. which was calculated in all cases where the farmer used a diesel engine, electricity, or solar as a power source for a drip irrigation system.

2.5.4. Social Analysis

This is a very simple analysis to find out the main reasons for the adoption of different power sources for drip irrigation systems. It was fully based on the farmers' consent and was evaluated by a questionnaire while seeking the interest of the farmer on the basis of simple questions. Based on the farmer's approach, whether he can afford it or not is easily approachable, as is his agricultural education and interest, interactions with the farming community, and so on, which was interpreted by using the SPSS model to run the chi-squared test and evaluate adaptation by the farming community in Punjab.

## 3. Results and Discussions

*3.1. Economic Viability of Different Power Sources for Drip Irrigation System*

To determine which source for drip irrigation system is more viable, three parameters were considered: benefit–cost analysis, payback period, and life cycle cost.

*3.2. Benefit–Cost Analysis of Different Power Sources for Drip Irrigation System*

The benefit:cost ratio for each power source was calculated by using Equation (1), where net present value was calculated by equation 3 for each source and site individually. All the benefit and expense costs were in US dollars.

As shown in Table 1, electric power DIS with the highest B-C ratio of 1.65. Diesel power DIS had the lowest B-C ratio of 1.45 due to high operating costs, and the B-C ratio of solar power drip irrigation systems was 1.55, which was second only to electric power DIS, as shown in Figure 4.

**Table 1.** B-C ratio of different power sources for DIS.

| Site Code | Electricity Power | Diesel Power | Solar Power |
|:---:|:---:|:---:|:---:|
| 1 | 1.64 | 1.55 | 1.73 |
| 2 | 1.78 | 1.59 | 1.86 |
| 3 | 1.76 | 1.41 | 1.65 |
| 4 | 1.58 | 1.60 | 1.57 |
| 5 | 1.55 | 1.39 | 1.60 |
| 6 | 1.47 | 1.48 | 1.48 |
| 7 | 1.64 | 1.49 | 1.41 |
| 8 | 1.77 | 1.29 | 1.39 |
| 9 | 1.54 | 1.35 | 1.42 |
| 10 | 1.79 | 1.45 | 1.35 |
| 11 | 1.57 | 1.21 | 1.45 |
| 12 | 1.56 | 1.39 | 1.42 |
| 13 | 1.66 | 1.46 | 1.56 |
| 14 | 1.70 | 1.54 | 1.31 |
| 15 | 1.80 | 1.45 | 1.55 |
| Mean | 1.65 | 1.44 | 1.52 |
| St. Dev | 0.11 | 0.11 | 0.15 |

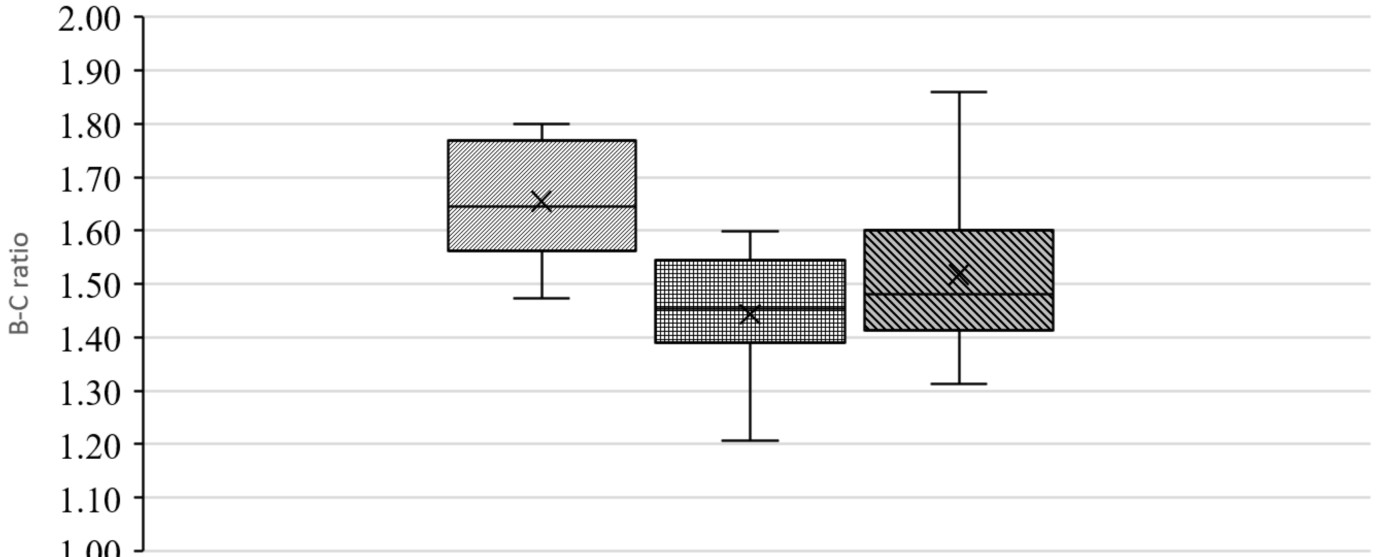

**Figure 4.** Benefit:cost ratio of different power sources for drip irrigation systems.

### 3.2.1. One-Way ANOVA

Reject the null hypothesis and accept the alternative hypothesis, which is that there is a significant difference between the benefit:cost ratio of different power sources. Table 2 shows that F = 11.215 and $p < 0.0001$.

**Table 2.** Results of One-Way ANOVA of B-C ratio for different power sources.

|  | Sum of Squares | df | Mean Square | F | Sig. |
|:---|:---:|:---:|:---:|:---:|:---:|
| Between Groups | 0.343 | 2 | 0.172 | 11.215 | 0.000 |
| Within Groups | 0.642 | 42 | 0.015 |  |  |
| Total | 0.986 | 44 |  |  |  |

To find the minimal difference between the means of B-C ratio for different power sources, we used a post hoc Tukey HSD test using the SPSS model.

There is a significant difference between the B-C ratio of electricity and diesel, solar and diesel and a nonsignificant difference between the B-C ratio solar and electric power drip irrigation system, which shows that solar and electric power systems are much better than the diesel power DIS presented in Table 3.

**Table 3.** Post hoc Tukey test for B-C ratio.

| (I) Factor | (J) Factor | Mean Difference (I-J) | Std. Error | Sig. | 95% Confidence Interval | |
|---|---|---|---|---|---|---|
| | | | | | Lower Bound | Upper Bound |
| Solar | electricity | −0.13733 * | 0.04516 | 0.011 | −0.2470 | −0.0276 |
| | diesel | 0.07333 | 0.04516 | 0.247 | −0.0364 | 0.1830 |
| Electricity | Solar | 0.13733 * | 0.04516 | 0.011 | 0.0276 | 0.2470 |
| | diesel | 0.21067 * | 0.04516 | 0.000 | 0.1010 | 0.3204 |
| Diesel | Solar | −0.07333 | 0.04516 | 0.247 | −0.1830 | 0.0364 |
| | electricity | −0.21067 * | 0.04516 | 0.000 | −0.3204 | −0.1010 |

Significant level at Alpha equal to 0.05 *.

### 3.2.2. Cashflow Diagrams for All Power Sources Used for DIS

Cashflow diagrams were constructed by calculating the means of all expenses as well as profit for all sites of each power source. It is easy to understand the range of every expense related to the system. Cashflows for all types of drip systems are presented in Figures 5–7.

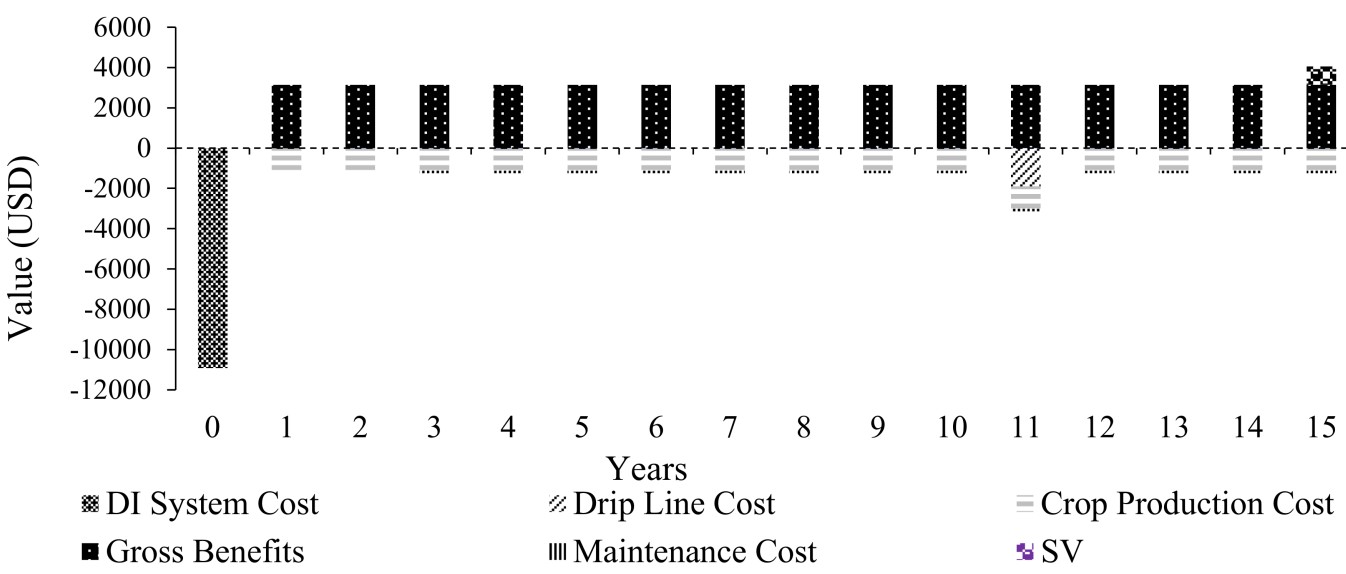

**Figure 5.** Cashflow diagram for solar power drip irrigation system.

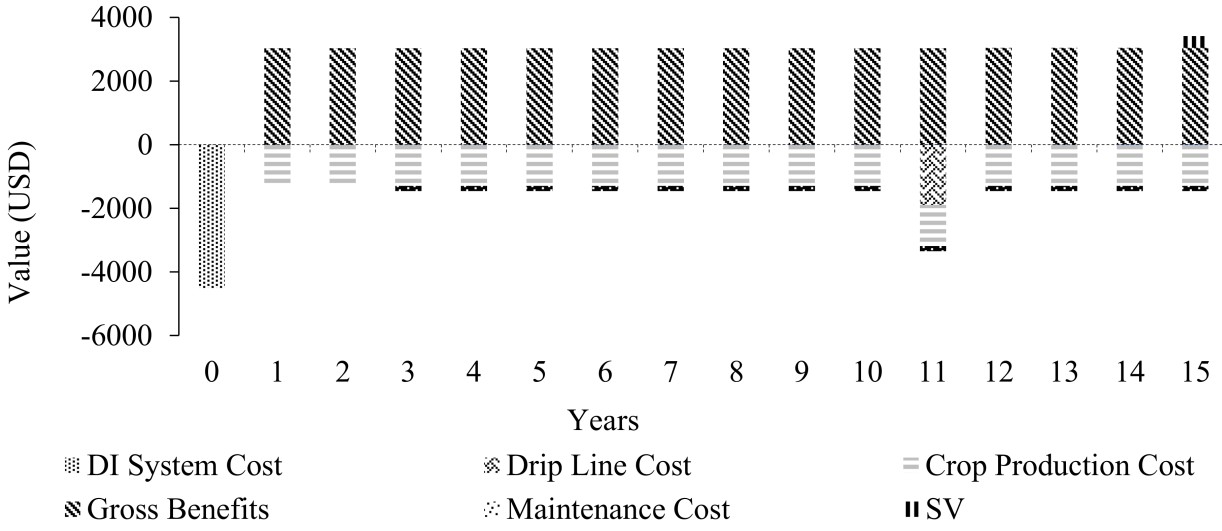

**Figure 6.** Cashflow diagram for electric power drip irrigation system.

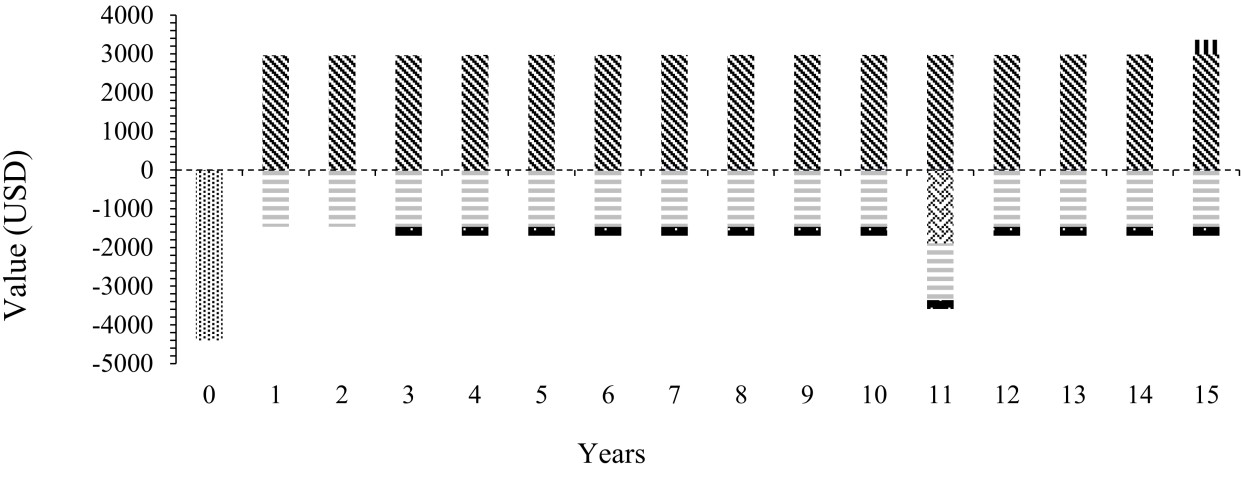

**Figure 7.** Cashflow diagram for diesel power drip irrigation system.

Cashflow Figure 7 describe all the system expenses, including salvage value, for each power source and maize crop production cycle, as well as gross benefits from a one-hectare area, which can be easily understood.

### 3.3. Payback Period of Different Power Sources for DIS

The payback period for each of the 15 drip irrigation sites for each source was calculated by using Equation (4). The project with a payback period of 3 years is mostly considered the best project.

Solar power DIS has the highest range of PBP, from 2 years to 7 years for different sites, with an average of 5.2 years, which is a very long duration for return of initial investment. Electricity has the lowest PBP, which ranges from 1 to 3 years with an average of 2 years. Electric power DIS has become the best power source with respect to PBP, as diesel power DIS sites also have a range of 1–4 years with an average of 2.5 years. Diesel power sources also have higher PBP than electric power DIS in Figure 8.

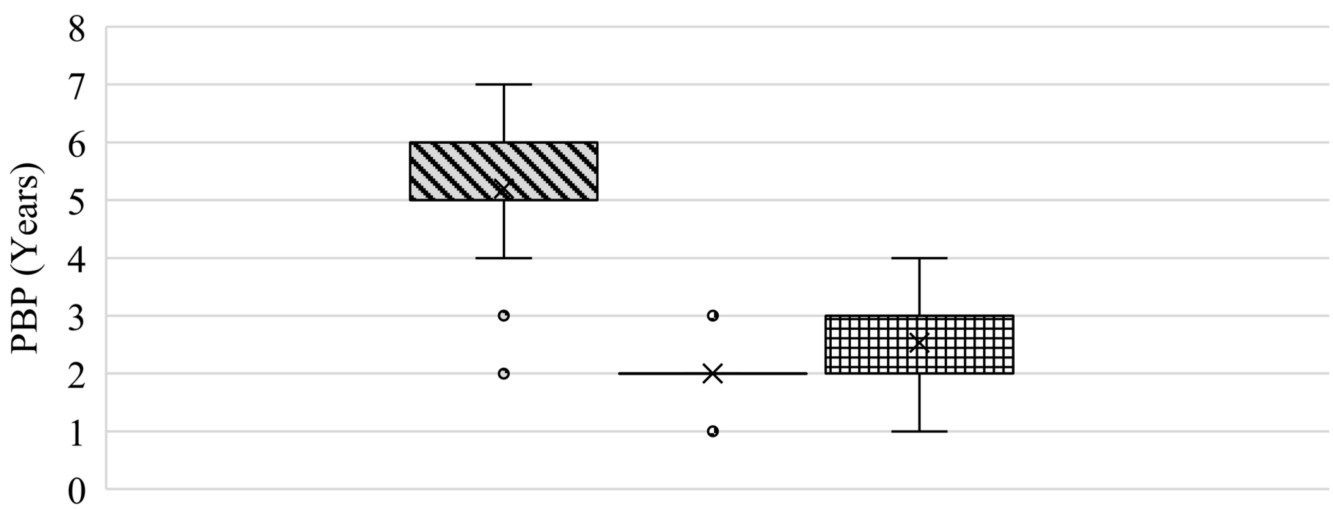

**Figure 8.** PBP of different power sources for DIS.

The average PBP as well as the variation to the average clearly shows that electric power DIS has the lowest PBP of 2 years compared to other power sources, as shown in Table 4.

**Table 4.** PBP of different power sources for DIS.

|  | Solar | Electricity | Diesel |
|---|---|---|---|
| Average | 5.2 | 2 | 2.53333 |
| St. Deviation | 1.373213 | 0.534522 | 0.99043 |

*3.4. Life Cycle Cost Analysis of Different Power Sources for Drip Irrigation System*

The life cycle cost was calculated for each power source by using Equation (5) for each source. All the calculations are made for an area of one hectare of maize crop for a period of 15 years, where the lateral is replaced after 10 years and the diesel engine after 15 years. All the costs are in US dollars with a depreciation rate of 15%. The life cycle cost from 15 sites for each power source for DIS is shown, along with the standard deviation. Diesel power DIS has the highest LCC at 13,628.47, whereas solar power DIS has 12,994.67 and electricity power DIS the lowest LCC of 9414.63. Electric power DIS is the best power source for decisions based on LCC analysis.

As shown in Figure 9, the life cycle cost of the diesel power drip irrigation system is higher than both the other power systems. It clearly shows that electric power drip irrigation systems have the lowest life cycle cost. Therefore, the best system we choose should have the lowest life cycle cost for the life period of 15 years. Figure 9 also shows the variation in LCC of solar power drip irrigation systems is much higher, which might be due to design parameters.

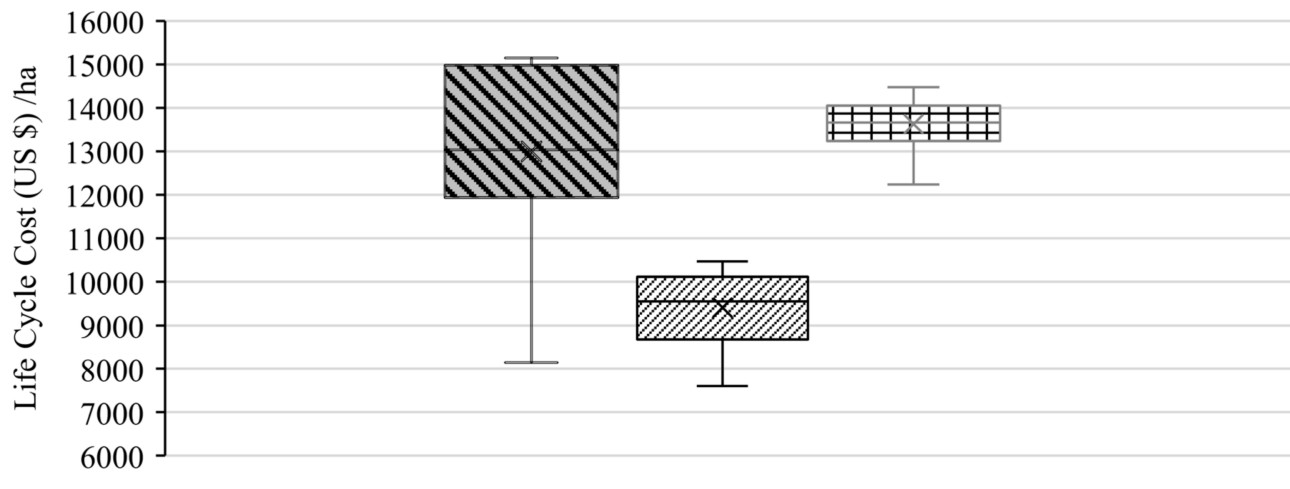

**Figure 9.** Life cycle cost of different power sources for drip irrigation systems.

Solar drip irrigation systems have the highest capital cost, which is approximately 6000 more than the other two sources, whereas diesel and electric power sources have nearly equal capital costs. Figure 10 also shows that diesel has higher maintenance and fuel costs, whereas solar power DIS has zero fuel cost. In the end, there is a nonsignificant difference in the LCC of solar and diesel power drip irrigation systems. Even diesel has a higher LCC than solar, which shows the capital cost of solar takes 15 years to be equal to the diesel power source.

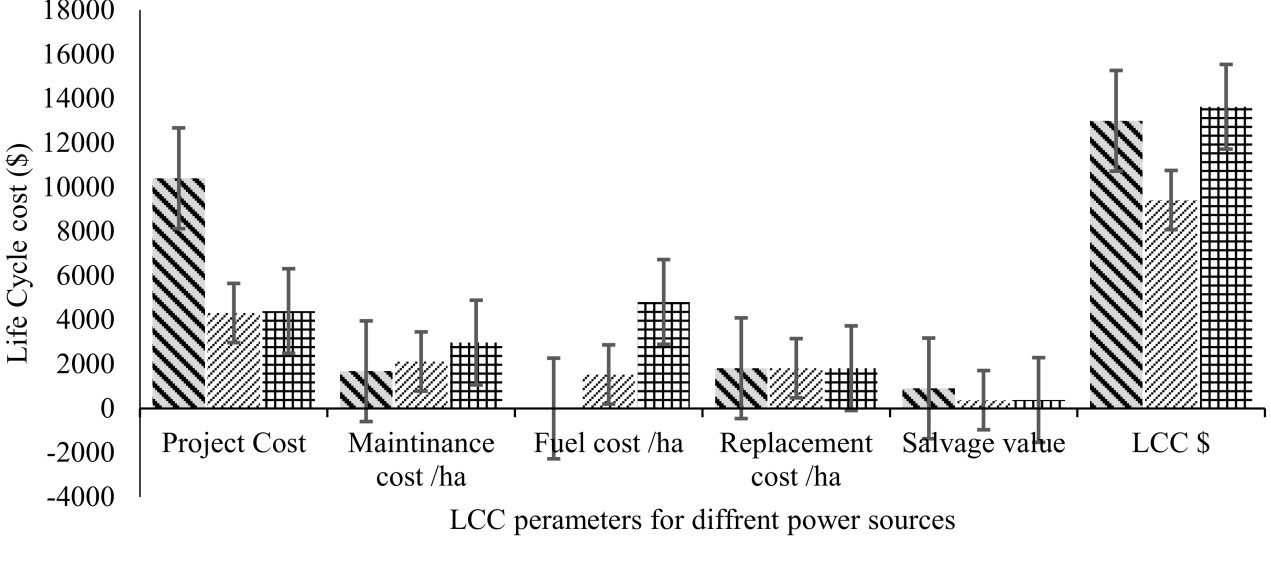

**Figure 10.** LCC parameters for different power sources for DIS.

According to the Ministry of Mines and Energy (2006), diesel engines need minor servicing four times a year for good-quality engines and two major services a year, minor services include oil and filter changes, whereas major services include decarbonization, oil change, and filter replacement by a skilled person. Furthermore, major and minor overhaul tasks require skilled people replacing crank shafts and drilling cylinders.

Production Cost for DIS

As shown in Table 5, the cost of production for maize was 0.21 USD/kg while using electric power DIS, which is a bit higher than diesel and solar power DIS, which were 0.173 USD/kg and 0.1729 USD/kg, respectively. By using an efficient irrigation system and limiting labor expenses, a progressive farmer in Bangladesh spends 1419 USD to produce 7.85 tons/ha of maize with a benefit:cost ratio of 1.42. Therefore, it can be easily calculated that they produce 1 kg of maize at a cost of 0.19 USD, which is almost the same as in Pakistan [43].

**Table 5.** Cost comparison of different power sources for DIS.

|  | Types of DIS by Power Source | | |
|---|---|---|---|
|  | **Solar** | **Diesel** | **Electricity** |
| LCC for drip irrigation system for 15 years | 12,994.67 * | 9414.629 * | 13,628.47 * |
| System cost for 1 year | 866.31 * | 627.64 * | 908.56 * |
| Mean production cost for DIS Production cost | 1118.15 * | 1296.28 * | 1467.64 * |
| Total cost | 1984.46 * | 1923.92 * | 2376.21 * |
| Mean crop benefits for DIS | 3130 * | 3027.61 * | 2965.14 * |
| Maize production (kg/ha) | 11,476.67 * | 11,101.23 * | 10,872.1 * |
| Production cost ($/kg maize) | 0.1729 * | 0.173 * | 0.21 * |
| Mean cost of production for maize crop by using DIS ($/kg) | | | 0.1853 * |

All calculation was done in USD *.

The progressive farmers of Panchagarh, while using an efficient irrigation system, produce maize at a rate of 6.35 tons/ha. Because of intensive management practice, they consider it a good yield. He calculated the production cost to be 0.15 USD/kg, which is less than what I calculated using HEIS. The dribble water system spares about 38% of water and improves yield by up to 55% as compared to the surface water system [44]. Attainability examined of solar power irrigation systems to save pastures in the inward magnolia of China, which had decayed because of atmospheric changes and overgrazing. Conditions were produced for investigation of the prime irrigation sum to structure the groundwater-based solar irrigation framework [45].

The real development of HEIS technology in China, however, occurred during the 1990s, and the area under microirrigation technologies increased to about 1.46 Mha in 2000, which further increased to around 3.00 Mha in 2005 and 4.59 Mha in 2010 [46].

*3.5. Environmental Impact of Different Power Sources for DIS*

For environmental impact assessment, $CO_2$ emissions were calculated for all power sources with the same sample and done by using emission factors for diesel and electricity via IPCC guidelines, and for solar power, the SEAP report was used for emission factor. Specifically, 74.1 (tco2/TJ), 0.613 (tco2/kW), and 0.02 (tco2/kW) [21].

The emission of $CO_2$ emissions ton/ha for diesel is much higher than other power sources, and solar power has a nominal amount of $CO_2$ ton/ha that is emitted for an area of one-hectare maize crop, as presented in Figure 11.

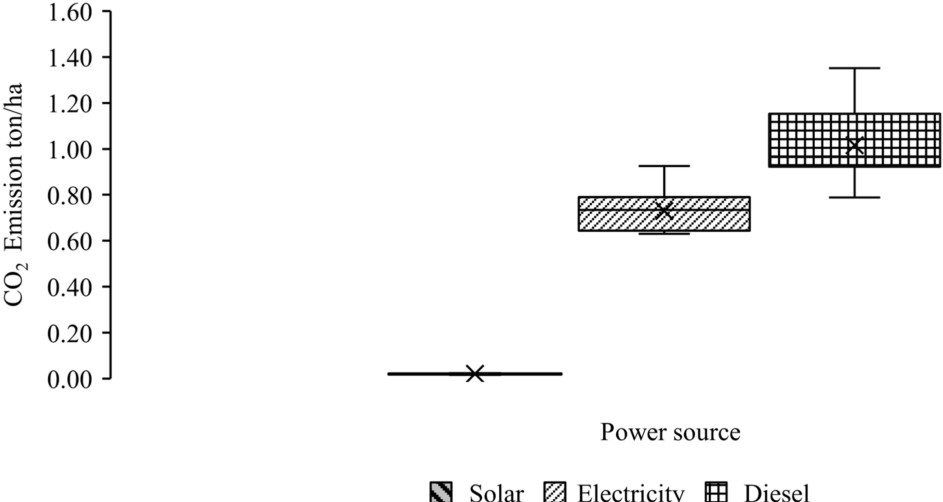

**Figure 11.** $CO_2$ emissions from different power sources for DIS.

From all the calculations and analysis, it was also calculated that diesel power DIS the diesel engine runs for 60.6 h to irrigate the maize crop and consumes about 323.40 L of diesel and emits 1.02 tons of $CO_2$/ha into the air, whereas electric power DIS runs for 56.13 h and consumes 1008.27 KW of electricity and emits 0.73 tons of $CO_2$/ha into the atmosphere, and solar power DIS consumes 1190.14 KW of electricity produced by solar and emits 0.02 tons of $CO_2$ during the life of the maize crop on ha area.

$CO_2$ Emission Form Each Source for 1 kg Production of Maize

Solar power DIS emits the least carbon dioxide, which is almost 0.0016 kg for the production of 1 kg of maize, whereas diesel power DIS emits 0.0848 kg, which is much higher than electricity and solar emits 0.0599 kg and 0.0016 kg, respectively, as presented in Figure 12.

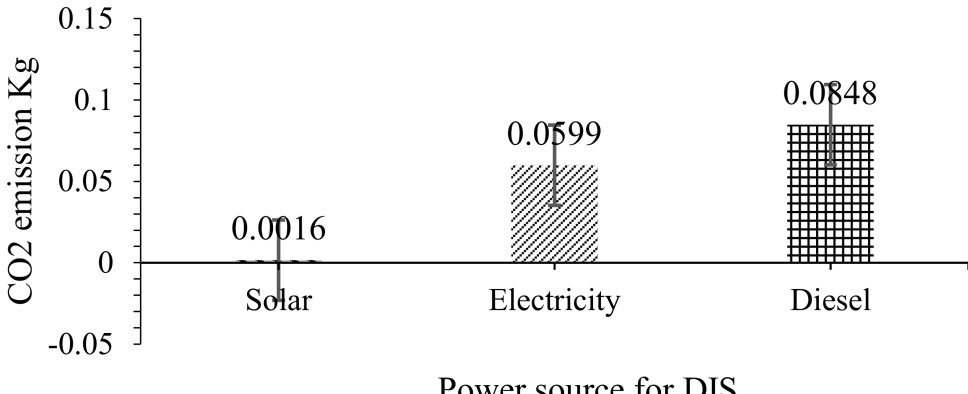

**Figure 12.** $CO_2$ emission kg/kg of maize production with different power sources for DIS.

*3.6. Social Adoption of Different Power Sources for Drip Irrigation System*

For social adoption, first the biodata of the farmer for each power source were collected, as given in the Table 6.

**Table 6.** Variables of the study for social adoption.

| Social Adoption Indicator | | Percentage % | | |
|---|---|---|---|---|
| | | **Solar** | **Electricity** | **Diesel** |
| Family members | 1–4 members | 20.00 | 13.33 | 6.67 |
| | 5–8 members | 46.67 | 53.33 | 60.00 |
| | 8–12 members | 33.33 | 33.33 | 33.33 |
| Total landholding | 5–3 ha | 40.00 | 60.00 | 60.00 |
| | 31–55 ha | 26.67 | 20.00 | 40.00 |
| | 56–80 ha | 33.33 | 20.00 | 0.00 |
| Education of head of family | Metric | 0.00 | 0.00 | 20.00 |
| | Intermediate | 0.00 | 0.00 | 13.33 |
| | Graduate | 6.67 | 53.33 | 53.33 |
| | Postgraduate | 93.33 | 46.67 | 13.33 |
| Education of farm operator | Illiterate | 20 | 40.00 | 66.67 |
| | Middle | 30 | 26.67 | 26.67 |
| | Metric | 50 | 26.67 | 6.67 |
| Occupation of HEIS owner | Farming | 26.67 | 26.67 | 53.33 |
| | Employee | 53.33 | 20.00 | 33.33 |
| | Business | 20.00 | 53.33 | 13.33 |
| Crop sown | Maize–potato | 40.00 | 66.67 | 60.00 |
| | Maize–potato–maize | 60.00 | 33.33 | 40.00 |
| Area under DIS | 1–3 ha | 100.00 | 100.00 | 100.00 |

The biodata of the farmer regarding his adoption of a power source show that 20% of farmers using solar power DIS have one to four family members. They also show that the farmers using solar power DIS are postgraduate and about 93% and 53% employees in the government and private sector, respectively, as presented in Table 6.

### 3.6.1. Head of Family Education

In sum, 93% of farmers with solar power DIS are postgraduates, with the remaining 7% being graduates. It is a much higher level of education than the farmers using diesel and electric power DIS. Farmers using electricity as a power source were also educated, with 47% postgraduates and 53% graduates. Farmers using diesel power DIS are less educated, as only 13.4% are postgraduate, while 54.4% are graduates, and the rest are intermediate and metric, which is much less education than the farmers of solar power DIS and electric power DIS.

### 3.6.2. Total Land under Control

It has been seen that farmers using solar power DIS have about 33% landholding of more than 55–80 ha, where farmers using electricity have only 20% in this range and diesel power DIS farmers have not even a single farmer in this range of landholding. On the other hand, among diesel and electric power DIS farmers, about 60% are under the limit of 5–30 ha of landholding, which results in farmers using solar power DIS having more landholding capacity than DIS farmers using electricity and diesel as power sources.

### 3.6.3. Occupation of DIS Owner

The occupation status for DIS farmers for each source shows clearly that farmers using solar power DIS are mostly employees in the government or private sector, which is about 53%, whereas in the other sources this is 20% for electricity and 33% for farmers of diesel power DIS, whereas 53% of electric power DIS farmers have a business and 54% of diesel power DIS farmers only farm. This means that the farmers using solar power DIS are mostly employees and well educated, whereas farmers using electricity

as a power source are mostly doing business and farmers using diesel power DIS are mostly doing farming.

### 3.6.4. Education of Farm Operators

Farm operators using solar power DIS are much more educated, with about 60% of them being metric, whereas the operators using electric power DIS have only 33% metric education. On the other hand, this is only 6% for diesel power DIS. About 67% of operators using diesel power DIS are illiterate.

All the farmers with drip irrigation systems without depending on a source are being subsidized by the government of Punjab. The farmers pay only 40% of the total cost for the whole system, except those using solar power DIS, who have paid an extra amount for the solar unit. This is also subsidized by the GOP, and the farmer pays only 20% of the total cost for the solar unit. The percentage of farmers wishing to adopt solar as a power source for drip irrigation systems at the next chance is 98%, whereas only 2% of farmers show willingness to adopt electric power DIS. These results show farmer interest and awareness of renewable energy.

Farmers using solar power DIS are much more active in their farming community, interacting with departments, the latest media, or other agriculture-related technologies than farmers using electricity or diesel power drip irrigation. On the other hand, diesel power DIS farmers have fewer interactions with the farming community, departments, and agricultural development media.

It is also shown that there is a significant difference between all power sources for interaction with agriculture and water management departments through different training. Such training gives information on the latest technologies and guides and trains farmers for their adoption and usage.

The farmer has the capability and interest to adopt solar power DIS because it is the need of the future, in addition to them having just a one-time cost to pay. Most of the farmers who live in cities and visit their farms every couple of days were more interested in this technology because they wanted to reduce their daily running costs to avoid any misappropriation, especially in the case of diesel engines. The DIS farmer requires daily diesel expenses and is dependent on the operator, whereas, as in the case of solar power DIS, there is no need for daily expenses. The number of laborers working at the farm may also be reduced. Solar technology, on the other hand, is prohibitively expensive for farmers to afford without government subsidies.

### 4. Conclusions

The B-C ratio of electricity power DIS is 1.65, which is higher than that of solar and diesel power DIS, which are 1.52 and 1.44, respectively. The LCC of diesel power sources is much higher than that of solar and electric power sources, and diesel has the lowest B-C ratio than of solar and electric power sources. DIS has the lowest LCC as well as the shortest payback period, which is 2 years, whereas diesel and solar have payback periods of 2.5 and 5.2 years, respectively. The production cost for maize is almost the same while using solar and electricity power DIS, i.e., 0.17 USD/kg of maize, where the high production cost is, i.e., 0.21 USD/kg while using diesel power DIS. Therefore, it can be easily concluded that electricity is the most viable source for DIS, but in the current energy scenario in Pakistan, electricity is not enough to meet the requirements of farmers. That is why we need to move to the second option, i.e., solar power DIS. It is much more costly, and farmers cannot afford this source. Diesel power DIS emits more $CO_2$ into the atmosphere, 1.016 tons/ha for the maize crop, whereas electricity emits 0.732 tons and solar emits 0.02 tons. Solar emits very little $CO_2$ when compared to other power sources. which suggests that the government should promote solar power DIS because it is more environmentally friendly than other power sources. Social benefits can also be increased by the adoption of solar energy for the operation of drip irrigation. The solar powered drip irrigation system was found to be a socially

feasible option on account of its impressive social benefit-to-cost ratio. This confirms that the wide adoption of drip irrigation will generate enough social benefits to justify the subsidization of drip irrigation in the country. Farmers using solar power DIS are more educated and have more interaction with the farming community and agriculture departments also. Maximum employment in solar energy DIS was also discovered. Diesel power DIS farmers have the lowest literacy rate and very limited interaction with the farming community and agriculture departments. Approximately 98% of farmers who use DIS indicate a willingness to use solar power DIS for their next project.

## 5. Recommendations

The government should take an initiative towards renewable energy, as it is environmentally friendly. Farmers also want to adopt it, but cannot because of the high capital cost. If the government subsidized solar as a power source for drip irrigation systems, more than 80% of farmers would switch to solar, which would also play a positive role in the electricity shortfall of the country, as we will save a lot of electricity from the agriculture sector by adopting solar as a power source. The solar power DIS should be used for high-value crops such as vegetables, maize, potatoes, and flowers for high economic returns, as the solar power drip irrigation system is a heavy investment indeed, which should return as soon as possible through cash crop adoption. Maintenance services should be provided by service and supply companies for at least five years, as the DIS payback period for solar power is at least five years. Drip irrigation system parts should be manufactured locally, and manufacturers should relax on extra taxes. It will lower the capital cost of solar power DIS, allowing farmers to easily adopt this system.

**Author Contributions:** Conceptualization, I.U.H. and M.N.; methodology, I.U.H. and T.M.; software, M.M.O.; formal analysis, M.Y. and M.Z.; investigation, I.U.H. and T.M.; data curation, M.Z. and M.M.O.; writing—original draft preparation, S.A. (Sikandar Ali) and M.Z.; writing—review and editing, S.A. (Shaheer Ahmad) and M.Z.; supervision, M.Y. and M.N. All authors have read and agreed to the published version of the manuscript.

**Funding:** This research received no external funding.

**Data Availability Statement:** The data are unavailable due to privacy restrictions.

**Acknowledgments:** The authors want to thank Asian Institute of Technology, Thailand and University of Agriculture, Faisalabad, Pakistan (UAF) for their support during this research.

**Conflicts of Interest:** The authors have no conflicts of interest.

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
