# Peer review of "Socioeconomic and Environmental Impact Assessment of Different Power-Sourced Drip Irrigation Systems in Punjab, Pakistan"

_agriengineering, doi:10.3390/agriengineering5010016_

Round 1

Reviewer 1 Report

The paper is well structured and has very interesting local information. However, it does not provide great novelty in contrast to what is already well known about the use of these different energies in drip irrigation.

The conclusions are obvious, the results in technical and environmental aspects are the expected, but present interesting facts in social aspect of Pakistan´s interested party.

The paper is interesting to disseminate a local case in Pakistan of comparative use of different energies in drip irrigation systems, including environmental and social aspects. For these reasons, I recommend accept it after minor revision.

To better the understanding of paper I want to do the follow suggestions and questions:

a) You should consider rewriting point 2.4.9 “Fuel Cost.” (Lines 289 – 292)

In my opinion, it does not make sense to use fuel cost when talking about the cost of electrical energy. I suggest using energetic cost of running.

It is not correct to use kW/h to express electric energy, but instead use kW-h (line 291). The energy is the power multiplied by the time of use.

b) The Material and Method section overexplains well-known variables. However, it is not very clear to me how you calculate the investment costs. How is the investment cost estimated if there is not electricity available and use this energy? Is it diesel or solar energy chosen when there is electricity available, or have they been implemented only when there was no electricity? I am missing slightly more information in this regard.

c) In section 3.2.2. “Cash Flow Diagrams for all power sources used for DIS” (Lines 369 – 379). Regarding figures 6,7 and 8.

·      The legends are hard to read, modify the figures or legends so that they can be better appreciated. In addition, not using the same scale to all of them induces error, I recommend also changing this.

·      All the cash flows appear the same, except for the System cost. Why is it the same in the DIS powered by diesel as it is in the one that uses electric power?

·      Does it make sense to put the same cash flows that do not depend on the energy system used or is it better to focus on those that generate a difference between them?

Author Response

Reviewer 1 :

Sr No

Suggestions/Comments

Remarks

1

The paper is well structured and has very interesting local information. However, it does not provide great novelty in contrast to what is already well known about the use of these different energies in drip irrigation.

This paper is a case study for the country to satisfies the adopters to go for drip irrigation based on Solar and Electric power instead of diesel operated engine

2

 You should consider rewriting point 2.4.9 “Fuel Cost.” (Lines 289 – 292)

In my opinion, it does not make sense to use fuel cost when talking about the cost of electrical energy. I suggest using energetic cost of running.

It is not correct to use kW/h to express electric energy, but instead use kW-h (line 291). The energy is the power multiplied by the time of use.

The recommended suggestion has been incorporated in the revised manuscript of the line 283-285. The point 2.4.9 has been rewritten.

The unit has been corrected and written in the line 285 of the revised manuscript.

3

The Material and Method section overexplains well-known variables. However, it is not very clear to me how you calculate the investment costs. How is the investment cost estimated if there is not electricity available and use this energy? Is it diesel or solar energy chosen when there is electricity available, or have they been implemented only when there was no electricity? I am missing slightly more information in this regard.

Dear reviewer, Pakistan is a developing country and nowadays we are having shortfall of electricity not only in villages, it’s happening in our big cities too. Therefore, most of our farmers are moving towards suitable solutions. In most of our villages we do not have even electricity and if we have its way expensive now.

4

In section 3.2.2. “Cash Flow Diagrams for all power sources used for DIS” (Lines 369 – 379). Regarding figures 6,7 and 8.

·      The legends are hard to read, modify the figures or legends so that they can be better appreciated. In addition, not using the same scale to all of them induces error, I recommend also changing this.

·      All the cash flows appear the same, except for the System cost. Why is it the same in the DIS powered by diesel as it is in the one that uses electric power?

·      Does it make sense to put the same cash flows that do not depend on the energy system used or is it better to focus on those that generate a difference between them?

All the legends have been modified in the figure 6,7 and 8. The figures have been modified and the suggestions has been incorporated in the revised manuscript.

Reviewer 2 Report

Reviewer’s comments on Manuscript No. agriengineering-2051349

Title

:

Modification Needed. Remove Region Name its limits the scope of the study

Abstract

:

Need modifications as suggested in manuscript and grammatical corrections.

Keywords

:

Need modifications as suggested in manuscript. Arrange in Alphabetical Order

Introduction

:

An introduction of a research paper should have the following in sequential order to make the paper acceptable.

1.         The basic idea of the topic of the paper

2.         Citation of similar works done by others quoting the references in proper format of the journal.

3.         Some modifications as suggested in manuscript.

Materials and Methods

:

Need modifications as suggested in manuscript with following suggestions

1.      For proper and authentic results there should be proper and authentic as well as standard method.

2.                  If there is no citation in the methodology, it shows that it has been invented by you.

3.      Revise this portion to make the paper acceptable. Use citation to denote the methods. Be precise and focused.

4.      Statistical analysis should be summed in one sub-heading.

Result and Discussion

:

Need modifications as suggested in manuscript with following suggestions

1.      The manuscript lacks in proper results and discussion.

2.      The discussion is explanation of your results in the light of similar works done by citing them to make your statement valid.

3.      Revise this portion to make your paper relevant. Some more recent references should be incorporated to strengthen the advocated facts.

Conclusion

:

Need modifications as suggested in manuscript with following suggestions

1.      The manuscript includes conclusion of that study, which is not carried out in the paper, modify it.

2.      Manuscript should include the conclusion based on the results of the study shown.

Author Response

Reviewer 2 :

Sr No

Suggestions/Comments

Remarks

1

Title: Modification Needed. Remove Region Name its limits the scope of the study

As this investigation is a case study for the selected regions, that’s why the name of the regions is incorporated in the manuscript

2

Need modifications as suggested in manuscript and grammatical corrections.

The abstract has been re-written, and the grammatical mistakes has been removed

3

Need modifications as suggested in manuscript. Arrange in Alphabetical Order

The keywords have been arranged in the required sequence and has incorporated in the revised manuscript line 37

4

An introduction of a research paper should have the following in sequential order to make the paper acceptable.

1.         The basic idea of the topic of the paper

2.         Citation of similar works done by others quoting the references in proper format of the journal.

3.         Some modifications as suggested in manuscript.

1.     The basic purpose of the study is to explain the operation cost of solar drip irrigation in the selected regions of the country, and this is now clearly incorporated in the introduction section of the revised manuscript line 125-134

2.     The use of previous work has been sited in the revised manuscript and are highlighted red

3.     This study belongs to the economic analysis so suggestion regarding adaptation of solar and electric operated drip irrigation system has been described in the revised manuscript.

5

Material Method: Need modifications as suggested in manuscript with following suggestions

1.      For proper and authentic results there should be proper and authentic as well as standard method.

2.                  If there is no citation in the methodology, it shows that it has been invented by you.

3.      Revise this portion to make the paper acceptable. Use citation to denote the methods. Be precise and focused.

1.     The recommended suggestion has been incorporated in the revised manuscript line 151.

2.     The use of citation where mandatory has been incorporated in the revised manuscript line 139-140. However this investigation is only relevant to provide information about the adaptation of drip irrigation via solar , electric operated

3.     The material and method have been modified. The sampling procedure has been mentioned now.

6.

Results and discussion: Need modifications as suggested in manuscript with following suggestions

 1.      The manuscript lacks in proper results and discussion.

2.      The discussion is explanation of your results in the light of similar works done by citing them to make your statement valid.

3.      Revise this portion to make your paper relevant. Some more recent references should be incorporated to strengthen the advocated facts.

The relevant and suggested recommendations have been added in the revised manuscript.

7

 Need modifications as suggested in manuscript with following suggestions

1.      The manuscript includes conclusion of that study, which is not carried out in the paper, modify it.

2.      Manuscript should include the conclusion based on the results of the study shown.

The conclusion portion has been incorporated in the revised manuscript line 539-564. And the concluded results have also been mentioned in the manuscript abstract line 32-35 and in the conclusion portion.

Reviewer 3 Report

This manuscript presents the evaluation and comparison of the economic viability, environmental impact, and social of different power sources for maize crops under drip irrigation systems in Punjab, Pakistan.

In general, the English language and style need to be improved by a native English speaker, for a better understanding of what is written.

The methodology is very well written, with clear and direct language, presenting enough details for the reproduction of the research.

Fig 4 should be deleted, once all the information presented already is shown in Fig 3.

Various times throughout the text, the authors confuse the sources with the numbers presented... but the mistakes identified were respectively discarded and commented directly on the attached PDF.

The authors should significantly improve the discussion of the results and interpret them based on recent and relevant references. In the Results and Discussion section, only four references are cited. The findings and their implications should be discussed in the broadest context possible, and the limitations of the work highlighted. I can say that this is the weakest point of the manuscript.

All tables and figures (except Fig 3) are properly cited in the text, as well all citations are listed in the references section. The equations aren’t cited.

The manuscript is suitable for publication after major revisions and adjustments which are pointed out in the manuscript revised file (see attached at the system).

Author Response

Reviewer 3 :

Page Number

Reviewer comments

Remarks

38

Keywords should arrange in alphabetical order

The suggestion has been incorporated in the revised manuscript in line 37

69, 70

Cubic meter should correct as m3

The suggestion has been incorporated in the revised manuscript line 50-51

99

It’s better to use CO2 instead of Carbon dioxide

The suggestion has been incorporated in the revised manuscript line 97

140- 144

Rewrite the objectives. Don’t use the following words to mention objectives. "First objective and secondly"

The suggestion has been incorporated in the revised manuscript line 130-134

173,182

Use one style to mention land size in whole text  (ha or hectare)

The suggestion has been incorporated in the revised manuscript line 160,170

309 -315

It’s better to mention what is the used version of SPSS

SPSS 26 and mentioned in line 301 of the revised manuscript

414

Mention figure number in main text

The suggestion has been incorporated in the revised manuscript of line 412

434 -462

It is unclear. It’s better to use two column  table to mention these equations

The suggestion has been incorporated in the revised manuscript

585, 586

It’s better to use the word “other option”  instead of the word second option

The suggestion has been incorporated in the revised manuscript

Keep a space between the value and the percentage mark. In some places, it is can be visible  and in some places, it is not

The suggestion has been incorporated in the revised manuscript

All formulas should be written in formula mode

The suggestion has been incorporated in the revised manuscript

Rearrange all tables and figures. There are few format errors.

The suggestion has been incorporated in the revised manuscript

Reviewer 4 Report

After careful reading, I consider that the structure, logical flow, literature review and statistics used in this manuscript are not up to the standards.

Authors made frequent mistakes throughout the MS. The authors would do well to refer to other peer-reviewed publications for guidelines on what is most appropriate in tables, results, and figures, and what is better placed in an appendix. Although I am aware that there is a great effort behind the manuscript, there still are several difficult parts for publication.

Page Number

Reviewer comments

38

Keywords should arrange in alphabetical order

69, 70

Cubic meter should correct as m3

99

It’s better to use CO2  instead of Carbon dioxide

140- 144

Rewrite the objectives. Don’t use the following words to mention objectives. "First objective and secondly"

173,182

Use one style to mention land size in whole text  (ha or hectare)

309 -315

It’s better to mention what is the used version of SPSS

414

Mention figure number in main text

434 -462

It is unclear. It’s better to use two column  table to mention these equations

585, 586

It’s better to use the word “other option”  instead of the word second option

Keep a space between the value and the percentage mark. In some places, it is can be visible  and in some places, it is not

All formulas should be written in formula mode

Rearrange all tables and figures. There are few format errors.

Author Response

(The authors gave the same response as above.)

Round 2

Reviewer 3 Report

The authors have significantly improved the manuscript; however, some points still need adjustments to make it suitable for publication.

The keywords need to be adjusted since words that already appear in the title should not be used as keywords.

I suggest using mm instead of inches as this is a more conventional unit for rain. 

In figure 2, note in the flowchart on the left where there are three times “15 sites for electric power”, I believe that one must be solar, another diesel, and the last electric.

In line 173 check the use of lowercase in the middle of a sentence.

Fig 4 should be deleted, once all the information presented already is shown at the Fig 3.

In the description of equation 2 in line 257, use “pt” instead of “Pt” to become according to the equation.

Line 358 – use a comma instead of a point in the middle of the sentence.

The caption of the Figure 8 is “Cash flow diagram for electric power drip irrigation system” but I believe that should be diesel.

Line 392 – the equation number is wrong.

Line 415 - use a comma instead of a point in the middle of the sentence.

Lines 428 and 429 - The values of the cost of production for maize to solar and electric power DIS are inverted.

Line 462 – The caption number should be 3.5.1

Lines 464 and 465 - Check these values and the text, as well, because it says that the CO2 emission of diesel is much higher than electricity and diesel... shouldn't it be solar?

Line 471 - Indicate the table number, and I suggest don’t use the term “below”.

So, the manuscript will be suitable for publication after minor revisions and adjustments which are pointed out in the manuscript revised file (see attached at the system) and listed in this section.

Author Response

Reviewer 3 Comments

Sr.no.

Line

Comments

Line

Answers

1

-

The authors have significantly improved the manuscript; however, some points still need adjustments to make it suitable for publication

-

Thank you so much for your appreciation, we have updated it accordingly.

2

37

The keywords need to be adjusted since words that already appear in the title should not be used as keywords

37

The recommended suggestion has been incorporated and highlighted in Dark Blue.

3

73-78

I suggest using mm instead of inches as this is a more conventional unit for rain

75-78

The recommended suggestion has been incorporated and highlighted in Dark Blue.

4

-

In figure 2, note in the flowchart on the left where there are three times “15 sites for electric power”, I believe that one must be solar, another diesel, and the last electric

The recommended suggestion has been incorporated.

5

173

check the use of lowercase in the middle of a sentence

173

The recommended suggestion has been incorporated and highlighted in Dark Blue.

6

-

Fig 4 should be deleted, once all the information presented already is shown at the Fig 3.

-

Figure 4 have been deleted on upon the recommendation of reviewer and figures number has been rearrange after the removal and highlighted in Dark Blue .

7

257

In the description of equation 2 in line 257, use “pt” instead of “Pt” to become according to the equation.

254

The recommended suggestion has been incorporated and highlighted in Dark Blue.

8

358

use a comma instead of a point in the middle of the sentence.

355

The recommended suggestion has been incorporated and highlighted in Dark Blue

9

370

The caption of the Figure 8 is “Cash flow diagram for electric power drip irrigation system” but I believe that should be diesel.

367

The recommended suggestion has been incorporated and highlighted in Dark Blue

10

392

the equation number is wrong.

373

The recommended suggestion has been incorporated and highlighted in Dark Blue

11

415

use a comma instead of a point in the middle of the sentence.

412

The recommended suggestion has been incorporated and highlighted in Dark Blue

12

428,429

The values of the cost of production for maize to solar and electric power DIS are inverted.

423-425

The recommended suggestion has been incorporated and highlighted in Dark Blue

13

462

The caption number should be 3.5.1

458

The recommended suggestion has been incorporated and highlighted in Dark Blue

14

464,465

Check these values and the text, as well, because it says that the CO2 emission of diesel is much higher than electricity and diesel... shouldn't it be solar?

461

The recommended suggestion has been incorporated and highlighted in Dark Blue

15

471

Indicate the table number, and I suggest don’t use the term “below”.

467

The recommended suggestion has been incorporated and highlighted in Dark Blue

Reviewer 4 Report

Authors did extensive editing and improvements than initial submission. Now, this manuscript is up to the stand for publishing. 

Author Response

Reviewer 4 Comments

Comment:

Authors did extensive editing and improvements than initial submission. Now, this manuscript is up to the stand for publishing. 

Answer:

Dear reviewer thanks a lot for your kind comment.